# Development of Reverse Transcription Recombinase Polymerase Amplification (RT-RPA): A Methodology for Quick Diagnosis of Potato Leafroll Viral Disease in Potato

**DOI:** 10.3390/ijms24032511

**Published:** 2023-01-28

**Authors:** Ravinder Kumar, Priyanka Kaundal, Rahul Kumar Tiwari, Milan Kumar Lal, Hema Kumari, Rakesh Kumar, Kailash Chandra Naga, Awadhesh Kumar, Brajesh Singh, Vinay Sagar, Sanjeev Sharma

**Affiliations:** 1ICAR-Central Potato Research Institute, Shimla 171001, Himachal Pradesh, India; 2ICAR-National Rice Research Institute, Cuttack 753006, Odisha, India

**Keywords:** sensitivity, simple RNA extract, potato, RT-PCR, RT-RPA, potato, DAS-ELISA, isothermal, PLRV

## Abstract

Potatoes are developed vegetatively from tubers, and therefore potato virus transmission is always a possibility. The potato leafroll virus (PLRV) is a highly devastating virus of the genus Polerovirus and family Luteoviridae and is regarded as the second-most destructive virus after Potato virus Y. Multiple species of aphids are responsible for the persistent and non-propagating transmission of PLRV. Due to intrinsic tuber damage (net necrosis), the yield and quality are drastically diminished. PLRV is mostly found in phloem cells and in extremely low amounts. Therefore, we have attempted to detect PLRV in both potato tuber and leaves using a highly sensitive, reliable and cheap method of one-step reverse transcription-recombinase polymerase amplification (RT-RPA). In this study, an isothermal amplification and detection approach was used for efficient results. Out of the three tested primer sets, one efficiently amplified a 153-bp product based on the coat protein gene. In the present study, there was no cross-reactivity with other potato viruses and the optimal amplification reaction time was thirty minutes. The products of RT-RPA were amplified at a temperature between 38 and 42 °C using a simple heating block/water bath. The present developed protocol of one-step RT-RPA was reported to be highly sensitive for both leaves and tuber tissues equally in comparison to the conventional reverse transcription-polymerase chain reaction (RT-PCR) method. By using template RNA extracted employing a cellular disc paper-based extraction procedure, the method was not only simplified but it detected the virus as effectively as purified total RNA. The simplified one-step RT-RPA test was proven to be successful by detecting PLRV in 129 samples of various potato cultivars (each consisting of leaves and tubers). According to our knowledge, this is the first report of a one-step RT-RPA performed using simple RNA extracted from cellular disc paper that is equally sensitive and specific for detecting PLRV in potatoes. In terms of versatility, durability and the freedom of a highly purified RNA template, the one-step RT-RPA assay exceeds the RT-PCR assay, making it an effective alternative for the certification of planting materials, breeding for virus resistance and disease monitoring.

## 1. Introduction

The potato (*Solanum tuberosum* L.) is regarded as the most important non-cereal crop in the world. The potato is believed to have its origin 8000 years ago in the high hills of the Andean Mountains of South America and it has become popular worldwide [1,2,3,4]. The United Nations have designated potatoes as a food that can potentially secure global food security [5,6]. Potatoes are a very nutritious crop and are regarded as a futuristic food for meeting present needs and can grow in almost any environment [7,8,9,10,11]. Potatoes are easier to cultivate and more productive than many other vegetables, and the harvested tubers can be further sown [12,13,14,15]. In 2020, the FAO estimated that more than 359 million metric tonnes of potatoes would be harvested from more than 17 million hectares of land. The present scenario of potatoes in India in terms of area, production and productivity was reported to be 2.20 million hectares, 53.58 million metric tonnes and 24.35 t/ha, respectively [12,16]. The potato is cultivated mainly for its quality tubers which are consumed in various forms of culinary items. However, various abiotic factors highly associated with the reduction in production and productivity have been suggested, including multiple types of fungi, bacteria and viruses capable of easily infecting potato crops. Viruses are the leading cause of seed potato degeneration and certification rejection. To identify these viruses in the lab and field conditions, sensitive, precise and cheaper methods must be developed which might be beneficial for the seed industry as well as farmers. Due to the high moisture content, tubers are frequently exposed to a variety of infectious viruses, with the potato leafroll virus (PLRV), potato virus Y (PVY), potato virus X (PVX) and potato virus S (PVS) being the most economically significant [16,17,18,19]. PLRV (genus Polerovirus, family Luteoviridae) has emerged as a global threat in healthy seed potato production systems among the more than 50 viruses that impact the potato crop globally [12,20]. PLRV has small, isometric virions that are predominantly confined to the phloem of infected plants (23–25 nm dia). It possesses a monopartite, single-stranded, non-enveloped genome with a 5882-nucleotide positive sense RNA. According to one estimate, this virus is responsible for 20 million tonnes of global crop loss [6,12,20,21]. PLRV activity is often indicated by the presence of symptoms such as upright and consistent rolling leaves, leathery leaves with progressive yellowing and reddening, significant plant height reduction, necrotic phloem tissue with prominent decaying spots sideways of leaf veins and ultimately necrosis in the potato tubers. However, the net necrosis of potato tubers depends upon the degree of infection and might increase during storage. A crimson or violet staining is visible on the leaf margins and undersides of certain cultivars. In general, potato leaf roll disease is transmitted by aphid vectors in a persistent, circulative and non-propagative manner [5,12,20,21]. However, pollen, seeds and mechanical inoculation are not transmission methods [12,18]. Virulent aphids from distant or neighbouring fields and unhealthy volunteer plants that develop from infected tubers generally cause initial infections. Infected potato plants will produce tubers infected with PLRV. If infected tubers are planted, contaminated plants will thrive. The green peach aphid is the most efficient and essential PLRV vector (*Myzus persicae*), but the virus is also carried by *Aphis gossypii*. *Macrosiphum euphorbiae*, which spreads to potato strains less successfully [21,22].

Although there are some impressive approaches to prevent disease, such as planting certified seeds that are virus-free or resistant to them [21,23,24,25], rouging diseased plants, administering insecticide to aphid populations and harvesting early, pesticide strategies have occasionally succeeded [20,26]. In this regard, one of the earliest methods used for detecting the virus is enzyme-linked immunosorbent assay (DAS-ELISA) [5,12,21]. To assess the virus status in potato seeds, there is a requirement for a cost-effective, dependable, rapid and highly sensitive protocol that can detect the virus in the early phase. Nevertheless, this process is expensive, time-consuming and often impracticable for use on dormant tubers [24,25,26,27]. Several alternative techniques have gradually been developed, including real-time reverse transcription-polymerase chain reaction (qRT-PCR) [27], nucleic acid sequence-based amplification [21,27], reverse transcription polymerase chain reaction (RT-PCR) [25,27], Northern blotting [27], Multiplex AmpliDet RNA [24,26] and immunocapture reverse transcription polymerase chain reaction (IC-RT-PCR) [21]. Recent studies have suggested that a visual (dye-based) detection of PLRV in infected leaves and tubers can be performed by employing the approach of reverse transcription loop-mediated isothermal amplification (RT-LAMP) [27,28]. Moreover, the recently developed method of unique single-temperature amplification has been used in recombinase polymerase amplification (RPA), which is gaining popularity in detecting and diagnosing plant pathogens [28,29]. During the RPA set-up, the prime constituent single-strand DNA binding protein (SSB) is utilized for appropriate strand stability. At the same time, two enzymes, recombinase and DNA polymerase, perform the strand displacement in the process of DNA amplification. The filament of recombinase protein formulates a compound with oligonucleotide primers, searches the target DNA for homologous sequences and induces restricted strand separation. The SSB stabilizes these split strands, while DNA polymerase expands the oligonucleotide priming sequences. The end product of RPA is the exponentially amplified template DNA at a uniform temperature (isothermal conditions) which can be easily visualized using a routine gel electrophoresis procedure. The products of RPA are immediately subjected to cloning followed by sequencing, similar to PCR, which is one of the major restrictions in the LAMP approach of diagnostics where the products are multimeric [28,29]. The minimal time requirement in an RPA setup and results generation is due to the modest primary denaturation of the double-stranded DNA (dsDNA) target. There are substantial studies that have previously documented the RPA-mediated detection of plant viruses. The technique has been successful in the detection of the banana bunchy top virus in several banana cultivars [30]. Likewise, the cardamom vein-clearing virus [31], cucumber green mottle mosaic virus [32] and cucumber mosaic virus in bananas [33] were intercepted and published using this RPA method. Even in potatoes, the potato virus S [34] and potato virus Y [35,36] have been detected with the RPA technique. Some other viruses which were successfully detected with this assay include the rose rosette virus [37], the maize chlorotic mottle virus [38], the piper yellow mottle virus [39] and the yam mosaic virus [40].

RNA extraction, on the other hand, is a difficult task that is regularly performed using several methods, each of which typically has disadvantages. In this study, simple RNA has been extracted from potato leaves and tubers using a nucleic acid extraction technique. In a recent study for other potato viruses (PVX and PVS), we examined three simple RNA isolation procedures to reduce the cost and make RPA easier [29,34,41]. By adding 0.75–1% sodium sulphite, the approach proposed by [41] was found to be highly successful for leaves and tubers. As crude RNA extraction in PLRV is difficult due to its phloem limiting factor and the high concentration of polyphenols and polysaccharides in potato leaves and potato tubers [29,34,41], sodium sulphite was added to a simple RNA extraction method [41]. Moreover, in our study, we examined the use of sodium sulphite which might affect the RNA yield from leaves and tubers using cellular disc paper. The developed method of one-step RT-RPA for detection of PLRV in potato leaves and dormant tubers might have a strong potential as a protocol in the field of diagnostics which uses the RNA extracted from cellular disc paper.

## 2. Results

### 2.1. Evaluation of Plant Samples and RT-RPA Primers for PLRV Detection

The three primers utilized in the current experiment successfully amplified the fragments of 249 bp (LRRPAF1/R1), 246 bp (LRRPAF2/R2) and 153 bp (LRRPAF3/R3), which was observed using cDNA by two-step RT-PCR under a thermal cycler in the PLRV of infected leaf samples (Figure 1 and Figure 2). After it was determined that each of the three primers worked properly in two-step RT-PCR, the two-step RT-RPA was carried out using cDNA at temperatures ranging from 36 to 42 °C and was tested using a thermal cycler (Figure 3). It was clear from the results that the PLRV can be identified satisfactorily in i RT-RPA at 36–42 °C. However, the primer set C (LRRPAF3/R3) was selected for further studies due to the shorter length of amplicon and performance as recommended by the manufacturers (Table 1).

### 2.2. Optimization of One-Step RT-RPA Method for the Detection of PLRV Using Optimized Primers

To make the protocol simple and cost-effective, one-step RT-RPA was standardized. PLRV was successfully amplified at temperatures of 38, 40 and 42 °C when using purified RNA as a template under isothermal conditions (heating block/water bath). However, when disc paper-based RNA was used as the template, out of four temperatures evaluated, no amplification was observed at 36 °C, while PLRV was successfully amplified at 38, 40 and 42 °C. In healthy controls, there was no amplification. It is clear from the results that the PLRV can be identified satisfactorily in one-step RT-RPA at 38–42 °C (Figure 4). Bearing the results in mind, a temperature of 40 °C was preferred for additional studies.

### 2.3. Detection of PLRV Using Selected Primers and Optimization of One-Step RT-RPA

To standardize the protocol experiments further, experiments regarding the minimum time requirement and optimum concentration were performed. An incubation time of 10 min was reported to be sufficient for the detection of PLRV in two-step RT-RPA; however, 20–30 min incubation was required for one-step RT-RPA, which can be seen in Figure 5. Subsequently, the appropriate concentration volume of magnesium acetate ranged from 12 to 20 mM; however, 14 mM was chosen for one-step RT-RPA optimization for better results.

### 2.4. Evaluation and Optimization of RNA Extraction Method

Although in a previous report [28] the cellular disc paper-based method was used to detect PLRV in leaves without the use of sodium sulphite salt (Figure 6), our results were conducted on a commercial kit. However, to improve RNA concentration and quality, an experiment was conducted to see the effect of different recommended concentrations of sodium sulphite on the reaction. The use of 0.75–1.0% sodium sulphite in the extraction buffer was reported to increase the quality of RNA extracted from leaves (Figure 6). In our report, it was suggested that the best amplification could be seen in the leaves in an extraction buffer of 1.0% sodium sulphite. A lower concentration of sodium sulphite (0.25–0.5%) also provided an amplification, but the band observed was faint or non-specific. However, in a further experiment, the use of 1.0% sodium sulphite and cellular disc paper was successful in the RNA extraction of potato leaves. However, the method was less sensitive in the detection of PLRV in tubers (data not shown).

### 2.5. Sensitivity and Specificity Analysis

For the sensitivity assay of one-step RT-RPA, the initial template of total RNA (1 μL) was subjected to a serial dilution ranging from 10^−0^ to 10^−8^. Subsequently, a comparative analysis with the previous protocols of RT-PCR was performed [25]. Additionally, simple RNA (1 μL) was also subjected to a serial dilution ranging from 10^−0^ to 10^−8^ for a similar comparative analysis. Agarose gel electrophoresis with concentrations of 2.5% (RT-RPA) and 1.0% (RT-PCR) revealed the limit of detection at various dilutions. We observed almost equivalent sensitivity of both the detection techniques/assay using primer LRRPAF3/R3 (Figure 7) in comparison with previously reported primers [12]. Furthermore, the limit of copy numbers using serially diluted plasmid was also determined which came to 3.27 × 10^6^ copies/µL (Appendix A).

Our study revealed that the interception of one-step RT-RPA in potato leaves was similar as compared with the RNA-bases RT-RPA and RT-PCR using a simple RNA extract. Therefore, the use of a modified simple RNA extraction technique was found effective in leaves when sensitivity was observed for one-step RT-RPA. There was no match observed with other viruses such as potato virus A (PVA), potato virus M (PVM), PVS, tomato Leafcurl New Delhi virus (ToLCNDV), PVX, PVY^NTN^, PVY^NT^, PVY^O^, groundnut bud necrosis virus (GBNV) and potato spindle tuber viroid (PSTVd) during the specificity analysis experiment (Appendix A).

### 2.6. Cloning and Sequencing of Amplicons for Primer Accuracy

The amplified target amplicons were extracted from the gel, then cloned and sequenced for accurate verification. The amplicon, which was isolated from the field, exhibited 100% sequence similarity with Indian PLRV isolates and other PLRV strains reported from Europe, Canada, Africa, America and the Czech regions. However, this 153 bp region was 98–99% similar to the Czech isolate and Spain isolate. The present results significantly validated the superiority and aptness of the RT-RPA assay. Interestingly, the other viruses were not detected in the specificity analysis experiments.

### 2.7. Validation of One-Step RT-RPA Results on Diverse Field Samples

For the study, 129 PLRV-suspected plant parts (asymptomatic and symptomatic) of widely cultivated and popular potato cultivars across different agroecological zones in various states in India were taken to test the optimized one-step RT-RPA method in the field (Table 2). As a healthy control, mother plants and mini-tubers that were grown from a contamination- and infection-free tissue culture facility were employed in this experiment. First, DAS-ELISA was used to test these 129 samples. Then, one-step RT-PCR and RT-RPA were used to confirm the results. Out of 129 samples of leaves, a random set of 15 samples was selected. The one-step RT-PCR detected thirteen samples as positive using earlier reported primers and primer set C using purified RNA in one-step RT-PCR and RT-RPA. However, one-step RT-RPA detected 12 samples as positive for PLRV when a simple RNA extract was taken as an RNA template (Figure 8).

Similarly, 10 tubers were tested randomly out of 129 tubers (dormant and stored) and a sharp band (positive) was observed in nine samples using previously reported primers and primer set C using purified RNA as a template in the one-step RT-PCR and RT-RPA, respectively (Figure 9). However, only two samples were detected as positive in one-step RT-RPA when disc paper-based RNA extract was used as a template (data not shown). The intensities of the amplicon (153 bp) of PLRV varied among samples which might be due to differences in the virus concentrations in the samples collected from diverse locations.

A detailed comparative performance analysis of DAS-ELISA, one-step RT-PCR and one-step RT-RPA on leaves, dormant tubers and sprouted tubers (in the case of DAS-ELISA) is shown in Table 3. For this activity, 27 samples from different states and cultivars were taken. Out of 27 samples of a different category, 13 leaf samples and 4 sprouted tuber samples tested positive for PLRV in the DAS-ELISA test. Through one-step RT-PCR, PLRV-positive leaf and tuber counts increased to 16 and 12, respectively. Here, purified RNA was used. Moreover, 26 leaves and 16 tubers were intercepted as PLRV positive using one-step RT-RPA coupled with purified RNA. Furthermore, when a simple RNA template was used, 26 leaves and 13 tubers samples were detected as positive with RT-RPA (Table 3). Overall, the modified one-step RT-RPA showed great promise through a hassle-free RNA extract, making it ideal for labs with limited resources to screen huge collections of wide and cultivated gene pools, field planting materials and robust virus screening experiments.

## 3. Discussion

During the course of this research, a reliable one-step RT-RPA assay was designed, and it was found to be suitable for the testing of PLRV on a wide scale in seed potato production. Seed tubers are the primary means by which viruses are disseminated. The fact that the established RT-RPA may be used for tubers confers further value of this work.

When compared to PCR, RT-PCR and ELISA tests [35,36,37], RPA has various advantages, the most notable of which are its rapid detection time and requirement for isothermal conditions (25–40 °C). In contrast to RT-RPA, the present approaches for detecting PLRV are labour-intensive, need highly specialized laboratory equipment and take a significant amount of time. Approximately 120 min are needed for RT-PCR (usually 35 cycles) before gel visualization may occur to obtain the desired result. Additionally, the DAS-ELISA assay for testing viruses needs a minimum of two days to complete. PLRV has also recently benefited from the introduction of the isothermal-based RT-LAMP [29,34]. However, the LAMP test requires a longer incubation time as well as four–six primers, which contributes to the expensive nature of the assay. The RT-RPA assay that was designed for PLRV can be carried out in a laboratory using a rough equipment, and it can deliver a detectable amplification in under half an hour without the utilization of a thermal cycler. In addition to that, the proposed test benefited from the utilization of cellular disc paper-mediated RNA extraction, which is relatively inexpensive. The current research has shown that a one-step RT-RPA using a simple RNA extract is just as successful as the more traditional RT-PCR technique. In comparison to RT-PCR from leaves, the sensitivity of the RNA preparation method based on cellulose disc paper was not affected in any way. This was accomplished by using 1.0% sodium sulphite in the extraction solution. In the past, it was stated that adding sodium sulphite to the extraction buffer created two to three times more RNA [29,34]. This was found to be the case here.

To achieve optimal primer specificity in RPA, it is necessary to analyse multiple different sets of primers [30,31,33,34,38]. Primers created for RPA are comparable to those designed for PCR; however, the lengths of the primers used in RPA are significantly greater [29,34]. In this experiment, we targeted the coat protein (CP) gene of PLRV due to the prevalence of this gene in the diseased potato [5,12,16]. The CP gene is also more apparent when reverse transcription is utilized because it is located at the 3’-end. The effectiveness of real-time reverse transcription polymerase amplification is significantly impacted by primer parameters, including annealing temperature, primer lengths and/or GC compositions [29,34]. During the process of designing the primers for PLRV detection with a one-step RT-RPA protocol, these aspects were given significant weight and consideration. As per RPA protocol, 30–35 bp primers were considered suitable for target amplification using RT-RPA. However, primers designed for standard PCR have been reported as equally good for RT-RPA-based virus detection [42]. The primers used in this study (27–31 bp) gave satisfactory results and the findings are concordant with previous successful reports [43,44,45]. A significant contribution to the development of specialized primers was made by the data on the genetic diversity of PLRV that is held at NCBI. This unique RT-RPA proved useful for detecting PLRV since the primers were designed from highly conserved portions of the coat protein gene of PLRV, which were available in the NCBI database.

The use of a one-step RT-RPA eliminates the necessity of performing separate phases of cDNA synthesis, which in turn lowers the expense of additional chemicals required for cDNA synthesis. Our methodology resulted in a considerable decrease in the amount of reaction volume required (11 µL as opposed to 50 µL) while maintaining the same level of sensitivity. Even though there is a larger upfront cost associated with RT-RPA chemicals, the overall cost of the test has been greatly decreased. Our findings are in agreement with a number of recently published papers [29,34,38,39] that have demonstrated the viability of RT-RPA using a relatively straightforward incubation apparatus.

The one-step RT-RPA assay that was developed for this work was put to the test on 129 field samples (one of each leaf and tuber), and when the results were compared to those of the standard DAS-ELISA and RT-PCR, they were found to be extremely encouraging. The number of positive samples found in leaves and tubers utilising the one-step RT-RPA detection approach was reported to be significantly higher than that of RT-PCR. The RT-RPA showed a greater sensitivity that was comparable to RT-PCR while also having a higher level of specificity, as shown in Table 2 and Table 3. Our conclusion is also consistent with findings from comparable studies that were conducted in the past using RT-RPA to detect PVX [29,34,39]. Additionally, a thorough comparison of different PLRV detection methods, such as DAS-ELISA, one-step RT-PCR [12] and one-step RT-RPA, through purified RNA and cellular disc paper-mediated RNA extract, showed that one-step RT-RPA was more effective in identifying the virus in leaves and tubers (Table 3). The evident variability in the detection of the virus from dormant and sprouted tubers was mainly due to uneven distribution ad low concentration of PLRV in dormant tubers. In sprouted tubers virus, the concentration increases up to the suitable detection range.

In contrast to the one-step RT-RPA method, the RT-PCR procedure is more vulnerable to the inhibitory chemicals that can be found in the extract of potato leaves and tubers. Additionally, a careful evaluation of various PLRV detection techniques, including DAS-ELISA, one-step RT-PCR [12] and one-step RT-RPA, using pure RNA and cellular disc paper-mediated RNA extract revealed that one-step RT-RPA was more successful in detecting the virus in leaves and tubers. This was accomplished with minimal sample preparation requirements. Our findings agree with the previously published sensitivity assay of RT-RPA for a variety of different plant viruses [24,30,31,33,34,35,36,37,38,39,40,41,46].

## 4. Materials and Methods

### 4.1. Source of Viral Pure Culture and Plant Materials

The infected potato plants were grown in the live virus culture maintenance facility at ICAR-Central Potato Research Institute (ICAR-CPRI) in Shimla, Himachal Pradesh, India. Both the RT-PCR (reverse transcription polymerase chain reaction) and the DAS-ELISA (double-antibody sandwich enzyme-linked immunosorbent test) employing a specific antiserum (Agdia Incorporated, Elkhart, IN, USA) demonstrated the presence of PLRV. Leaf or sprouting tissue was carefully crushed and diluted with 1:10 (*w*/*v*) in PBS-Tween buffer to conduct the DAS-ELISA (pH 7.4). The plates were then kept at 37 °C for two hours after the antiserum had been diluted to a ratio of 1:200 (*v*/*v*) in a buffer for the coating that had a pH of 9.6. After incubation, PBS-Tween was used to clean the plates and sap extract was added to each plate well. After that, the plates were heated to 37 °C for two hours, and then they were given a second washing with PBS-Tween. After incubating at 37 °C for two hours, alkaline phosphatase-antiserum was diluted to 1:200 in PBS-Tween. After the final wash, 1 mg/mL of p-nitrophenyl phosphate disodium was added to the substrate buffer. After incubation for one hour at room temperature, an ELISA reader was utilized to check for any responses at a wavelength of 405 nm (Tecan Sunrise^TM^ Männedorf, Switzerland). The values of optical density considered to be positive were those that were greater than three times the background level on average. The average background levels were determined for each ELISA plate by conducting tests on at least two wells containing all of the reagents except the sap extract. As a healthy control for the experiment, plants that had been grown from virus-free tissue cultures were utilized. Nuclease-free water was used in place of template RNA to conduct the non-template control, also known as the negative control.

### 4.2. The Process of RNA Isolation and cDNA Synthesis

Total RNA was isolated from both PLRV-infected and healthy plant sections using the Spectrum^TM^ kit and the manufacturer’s instructions. An amount of 2 μL of total RNA (approximately 100–200 ng/μL) or 2 μL of cellular disc paper-based RNA extract [28] was used in the first-strand cDNA synthesis experiment (Revert aid^TM^ First-strand cDNA synthesis kit, Life Technologies, USA). An amount of 25 mg of leaf tissue was crushed in 1 mL of extraction buffer #2 (800 mM guanidine isothiocyanate, 50 mM Tris-HCl pH 8.0, 0.5% (*v*/*v*) Triton X-100, 1% (*v*/*v*) Tween 20) for cellular disc paper-based RNA extraction. In a 0.2 mL PCR tube, a 3 mm-diameter disc of Whatman Grade 1 filter paper was placed. Then, 100 μL of leaf tissue homogenate was added and the tube was left at room temperature for 5 min before the mixture was removed and discarded. To clean the cellular disc, 200 μL of wash buffer (10 mM Tris-HCl buffer, pH 8.0, with 0.1% (*v*/*v*) Tween-20) was utilized. The whole RNA was then eluted at 94 °C for 5 min and kept at −80 °C in 35–50 μL of RNase-free water for further use. Using a Thermo ScientificTM NanoDrop 2000, the quality and quantity of cDNA and RNA were assessed (Thermo Fisher Scientific, Waltham, MA, USA). For cDNA, the absorbance ratio of 260/280 was 1.8–1.9, whereas, for RNA, it was 2.0–2.1.

### 4.3. Configuring the Reaction for One-Step RT-PCR

An Applied Biosystems GeneAmp9700 PCR system was used to perform the RT-PCR-based PLRV detection using a method described previously [25]. The 452 bp region around the PLRV coat protein gene was the focus of the RT-PCR reaction. A sensitivity analysis with previously reported primers [12] and similar annealing temperature conditions of the one-step RT-PCR was performed (Verso 1-Step RT-PCR Kit-Thermo Fisher Scientific, USA). The components were reverse transcriptase (0.5 μL, 200 of U/μL) and 0.5 μL of RNase Inhibitor. The RNA template consisted of 1 μL of either total RNA (100–200 ng/μL) or cellular disc paper-based RNA preparation. In one-step RT-PCR, the reaction mixture was made up of 1.0 μL of verso enzyme mix, 25 μL of 2X 1-step PCR ReddyMix, 2.5 μL of RT enhancer, 1.0 μL each of 10 M forward and reverse PCR primers, 1.0 μL of RNA and double-distilled water to make it up the final volume. The reaction profile was set to 50 °C for 15 min, followed by verso inactivation at 95 °C for 2 min, 35 cycles of 95 °C for 20 s, 30 s annealing temperature (58 °C), 1 min extension at 72 °C and a final extension at 72 °C for 5 min. The amplified products were run through an electrophoresis process in 1% (*w*/*v*) agarose gel stained with ethidium bromide solution in 1X TAE buffer at 100 volts for 1 h, along with a DNA ladder, and could be seen under UV light.

### 4.4. The Designing and Synthesis of Primers

Triplicate primer sets were designed for an efficacious one-step RT-RPA assay using (https://www.bioinformatics.nl/cgi-bin/primer3plus/primer3plus.cgi (accessed on 30 June 2021)) by following TwistAmp^®^, Aylesford, UK, reaction kit’s instructions (https://www.twistdx.co.uk (accessed on 30 June 2021). Based on sequences found in the National Center for Biotechnology Information (NCBI) database (http://www.ncbi.nlm.nih.gov/blast (accessed on 30 June 2021), the conserved region of the complete coat protein gene was chosen for making primers (AF453388, EU717546, AY307123, X77324, JQ420903, LC501445, FJ481109, MK445319, MH937415, MF062487, KR051205, KP090166, EF063711, D13954, KC456052, MG356502, KU586454 and GU256062). Figure 1 shows where each of these sequences fits in. Bio-Edit Software was used to align the sequences, and the nucleotide sequences that stayed the same were chosen. The Basic Local Alignment Search Tool (BLAST; http://www.ncbi.nlm.nih.gov/blast (accessed on 30 June 2021) was used to look at the specificity of primers in a workstation. All three primer sets (set A-LRRPAF1/R1, set B-LRRPAF2/R2 and set C-LRRPAF3/R3; Imperial Life Sciences (P) Limited, India) were tested to see how well they worked with the thermal cycler for RT-RPA (Figure 2). Table 1 shows a detailed list of all the primers, as well as the temperature ranges, GC content (%) and expected size of the products of amplification.

### 4.5. Optimization of RT-RPA and Primer Selection

The three primer sets (A, B and C) were employed in this investigation to search for PLRV, and the cDNA served as a template for RT-RPA. Based on its sensitivity and specificity, the primer set C (LRRPAF3/R3) was chosen for future research and the precise detection of PLRV. First, the freeze-dried reaction pellets were rehydrated with rehydration buffer (29.5 µL) and 8.2 µL nuclease-free water from the TwistDx, TwistAmp^®^ basic kit. This was necessary to kickstart the reaction. Additionally, to minimize the cost of reaction set-up, five reactions of 7.54 µL each from the initial reaction mixture were made in separate PCR tubes. Then, five reaction tubes were added with 1 µL of cDNA along with 10 pmol of forward and reverse primers (0.5 µL of each) and then 0.5 µL of magnesium acetate was added to each tube. The mixture was heated at 40 °C for 4 min on equipment such as a heating block, water bath or thermal cycler. The mixture was again mixed with a pipette and kept on the same apparatus. After 30 min of incubation, the reaction was stopped by keeping the tubes at 65 °C for 10 min. The visualization on a 2.5% gel gave the desired results. The single best primer was selected for subsequent experimentation only after visualization of the amplified product at different temperature regimes (36, 38, 40 and 42 °C).

### 4.6. The Optimization of RT-RPA in One Step for the Detection of PLRV

To make the reaction even easier, 0.5 μL of reverse transcriptase (200 U/μL), 0.5 μL of RNase Inhibitor (20 U/μL) and 1 μL of total RNA as a template were added to a one-step RT-RPA. The rest of the components were the same as the ones in the RT-RPA with cDNA, which was described in the section before this one. RNA was used as a template and RT-RPA was performed in one step at different temperatures (36, 38, 40 and 42 °C). Based on the results, the next set of tests was performed at 40 °C. In RT-RPA reactions, the best incubation time (10, 20, 30, 40 and 50 min) was also analysed. Then, for further experiments, an incubation time of 30 min was chosen for one-step RT-RPA. Standardization was also performed on the amounts of magnesium acetate (12, 14, 16, 18 and 20 mM). All of the tests were performed three or four times. A healthy control was also used to see if the RT-RPA showed any non-specific amplification.

### 4.7. Evaluation and Improvement of RNA Extraction Techniques in Leaves

As crude RNA extraction in PLRV was difficult due to its phloem limiting factor and the high concentration of polyphenols and polysaccharides in potato leaves and potato tuber, we tested earlier reported simple RNA isolation procedures to reduce the cost and make RPA easier [29,34] to make a low-cost one-step RT-RPA for PLRV. This method was compared with an RNA isolation method based on the Spectrum^TM^ Plant Total RNA kit (Sigma-Aldrich, St. Louis, MO, USA). We investigated several concentrations of sodium sulphite (0.25, 0.5, 0.75 and 1%) in simple RNA isolation from potato leaves and tubers to increase the quality of the RNA because the method itself was not very good at obtaining RNA from leaves and dormant tubers in the case of PLRV. To compare RT-PCR- and RT-RPA-based PLRV detection, purified RNA from the Spectrum^TM^ Plant Total RNA kit (Sigma-Aldrich, St. Louis, MO, USA) was also employed. The RNA preparation technique utilizing cellular disc paper was used to detect PLRV in potato leaves in light of the findings.

### 4.8. Analysis of the Sensitivity and Specificity of One-Step RT-PCR/RT-RPA

For the sensitivity assay of one-step RT-RPA, the initial template of total RNA (1 μL) was subjected to serial dilution ranging from 10^−0^ to 10^−8^. Subsequently, a comparative analysis with previous protocols of RT-PCR was performed [12]. Additionally, simple RNA (1 μL) was also subjected to a serial dilution ranging from 10^−0^ to 10^−8^ for a similar comparative analysis [29,34]. Depending on the situation, either the basic RNA extract or the healthy whole RNA was employed to create the serial dilutions. In addition, an assay was performed to analyse the limit of copy number detection using the serially diluted cloned plasmid (target) as a template in RPA for validation of primers sensitivity. Using the concentration and total length of the recombinant plasmid, its copy number was calculated using a dsDNA copy number calculator (https://cels.uri.edu/gsc/cndna.html (accessed on 5 June 2022). The initial copy number of the plasmid was 3.276 × 10^10^. A one-step RT-RPA assay was used to test the specificity of the method utilizing RNA taken from PVA, PVM, PVS, PVX, PVY^NTN^, PVY^NT^, PVY^O^, GBNV and PSTVd infected plants as well as a healthy control plant, and DNA isolated from ToLCNDV infected plants. Gel electrophoresis on agarose gels with concentrations of 2.5 and 1.0% for one-step RT-RPA and RT-PCR, respectively, was used to determine the detection limit.

### 4.9. Elution, Cloning and Sequencing for Specificity Confirmation

In order to conduct a specificity analysis, an RT-RPA amplification using the designed primers was carried out. Using a GeneJET^TM^ Gel Extraction Kit (Fermentas, Thermo Fisher Scientific, Waltham, MA, USA), the visualized amplicons on the agarose gel were extracted and then further cloned in pJET1.2/blunt plasmid vector (CloneJET PCR cloning kit, Fermentas, Thermo Fisher Scientific, Waltham, MA, USA) in *Escherichia coli* JM107. Following the confirmation of the positive clones by the colony PCR, sequencing was carried out on an ABI PRISM^TM^ 310 genetic analyser (Applied Biosystems-Hitachi, Foster City, California, USA) using M13 forward and reverse primers. Using BLAST, which can be found at http://www.ncbi.nlm.nih.govt/blast (accessed on 25 November 2022), the sequences that were obtained were analysed in the NCBI database.

### 4.10. Natural Infection Validation Using a Single RT-RPA Technique

The robustness of the optimized one-step RT-RPA method was further validated by testing 99 PLRV suspected field samples (leaves and tubers) of popular potato cultivars during the year 2019–2020. These samples were collected from fields across the states of India. The states of India that are believed to have contributed the suspect samples are as follows: Bihar, Gujarat, Haryana, Himachal Pradesh, Madhya Pradesh, Meghalaya, Punjab, Uttar Pradesh and West Bengal. Virus-free tissue culture-grown mother plants and mini-tubers were used as the samples for determining the overall health of the population. The validation procedure also made use of the tubers that had been saved from diseased plants. In addition to examining dormant tubers (without sprouts), sprouts and leaves, a comparison between DAS-ELISA and one-step RT-PCR/RT-RPA was conducted. Comparable results from this investigation were discovered (Table 2 and Table 3). The RT-PCR and DAS-ELISA procedures were carried out in the same way as previously described [12].

## 5. Conclusions

In order to detect PLRV in leaf and tuber samples, a one-step RT-RPA assay that is efficient and simple to use, in addition to being highly sensitive and specific, has been devised. The established technique can be carried out using only a small quantity of rapidly extracted simple RNA from fresh or preserved materials; purified RNA is not necessary for the analysis of leaf samples. In summary, the established method can replace conventional RT-PCR in several situations, including the screening of micro-propagated plants, disease-free planting material certification programmes and the screening of gene pools for PLRV resistance. Regarding the topical goal of resource optimization, this technique of PLRV detection utilizing simple RNA extracts of leaves and tubers followed by a single step of RT-RPA is promising and, to the best of our knowledge, is being documented here for the first time.

## Figures and Tables

**Figure 1 ijms-24-02511-f001:**
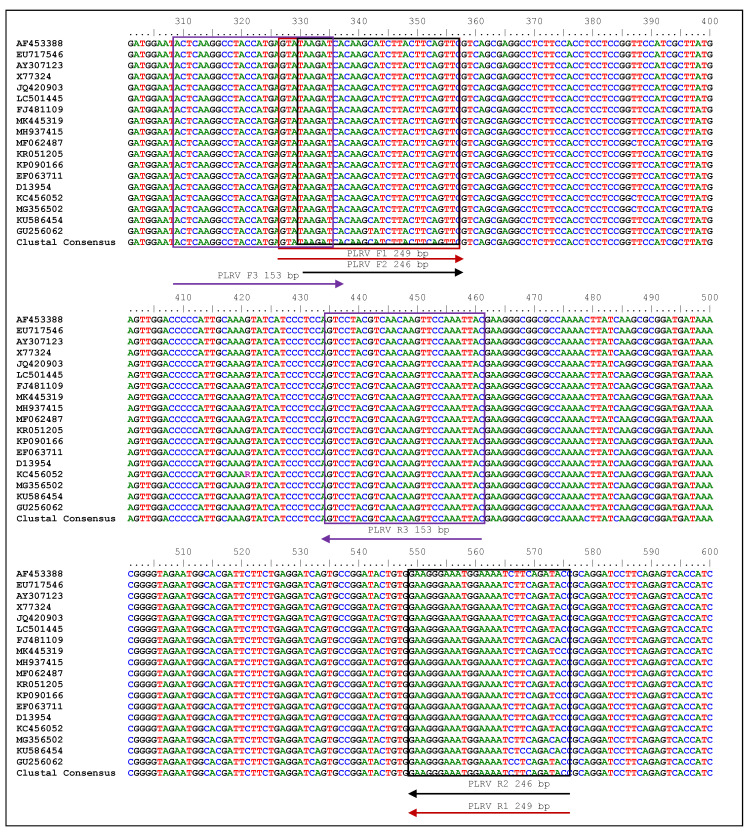
Location of primers in the potato leafroll virus (PLRV) coat protein gene. Free web-based Bio-edit software (https://www.bioinformatics.nl/cgi-bin/primer3plus/primer3plus.cgi/ (accessed on 30 June 2021)) was used to run a ClustalW multiple sequence alignment. The present study followed the manufacturer’s instructions, and primers in triplicate were generated using a full-coat protein gene by specifically targeting conserved regions. NCBI’s GenBank database (http://www.ncbi.nlm.nih.gov/blast (accessed on 30 June 2021)) provided the sequences (Accessions AF453388, EU717546, AY307123, X77324, JQ420903, LC501445, FJ481109, MK445319, MH937415, MF062487, KR051205, KP090166, EF063711, D13954, KC456052, MG356502, KU586454 and GU256062). The boxes denote primer regions, while arrows have been placed to specify the primer positions for reverse transcription-recombinase polymerase amplification (RT-RPA).

**Figure 2 ijms-24-02511-f002:**
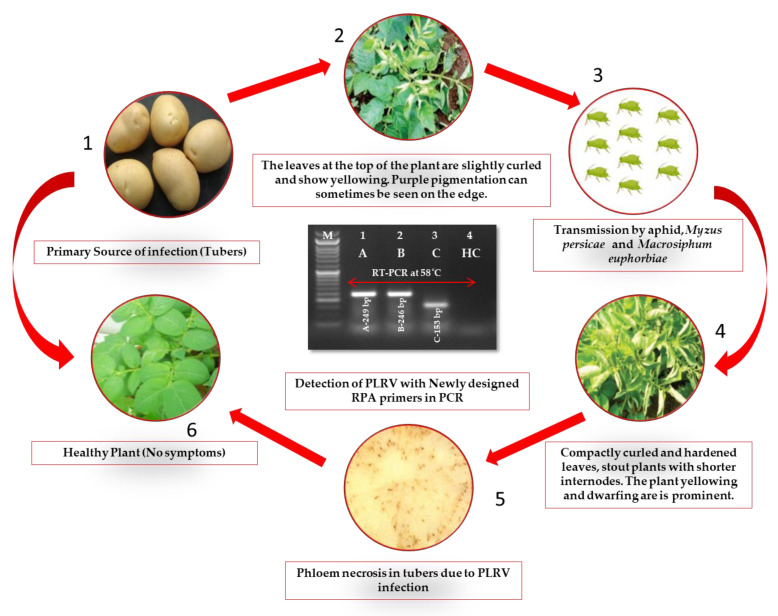
Illustrative model depicting the potato leafroll virus (PLRV) dissemination, symptoms of severe infection in known samples, and subsequent evaluation of virus by reverse transcription-polymerase chain reaction (RT-PCR) via newly designed primer sets in this investigation. Under current experimentation, PLRV symptomatic leaf samples were procured from a pure virus culture maintenance facility at ICAR-Central Potato Research Institute (ICAR-CPRI), Shimla, Himachal Pradesh, India. The cDNA synthesis was executed with 2 μL of 200–400 ng/μL total RNA with subsequent RT-PCR at 58 °C. The gel documentation-generated image denotes standard 100 bp ladder (Lane M), three primers set (LRRPAF1/R1: Lane 1, LRRPAF2/R2: Lane 2 and LRRPAF3/R3: Lane 3), along with the healthy control (Lane 4). The expected amplified products (bp) of 246 bp, 249 bp and 153 bp are also denoted in each lane for respective primers.

**Figure 3 ijms-24-02511-f003:**
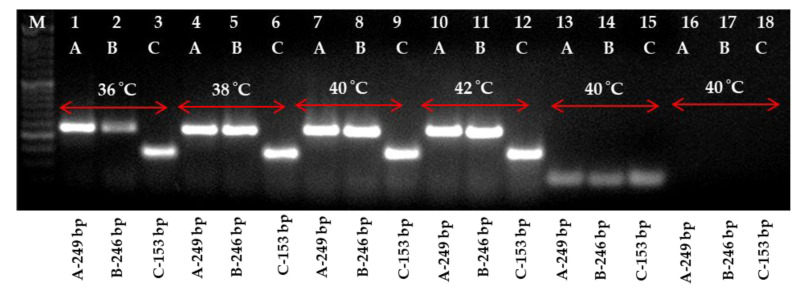
Primer optimization at different temperature regimes for two-step reverse transcription-recombinase polymerase amplification (RT-RPA) detection in potato leafroll virus (PLRV) infected samples. A total of 18 lanes depicted the respective target product amplification employing forward and reverse primers, viz., LRRPAF1/R1, LRRPAF2/R2 and LRRPAF3/R3. The primers sets are shown over each lane along with the corresponding expected amplified product. A 100 bp ladder (Lane M) along with a healthy control (Lanes 13, 14 and 15) and a negative (water) control (lanes 16, 17 and 18) are also included for the accuracy of the results.

**Figure 4 ijms-24-02511-f004:**
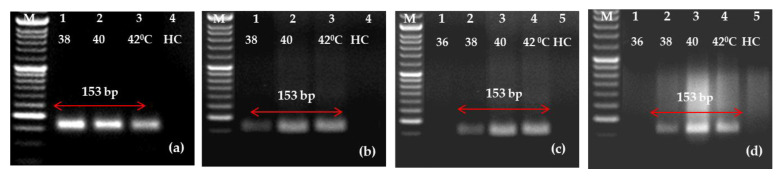
Optimization of one-step reverse transcription-recombinase polymerase amplification (RT-RPA) for potato leafroll virus (PLRV) detection using primer pair C (LRRPAF3/R3) at different temperature regimes (38, 40 and 42 °C). The gel-electrophoresis generated images depict the RT-RPA amplified amplicons in a heating block (**a**,**c**) and in a water bath (**b**,**d**). The purified RNA as the template was used in RT-RPA in (**a**,**b**); whereas the simple RNA has been used for (**c**,**d**). Within each gel documentation image, three lanes (1, 2 and 3) depict the PLRV-infected samples along with a 50 bp ladder (Lane M). Lane 5 (**c**,**d**) denotes the healthy control.

**Figure 5 ijms-24-02511-f005:**
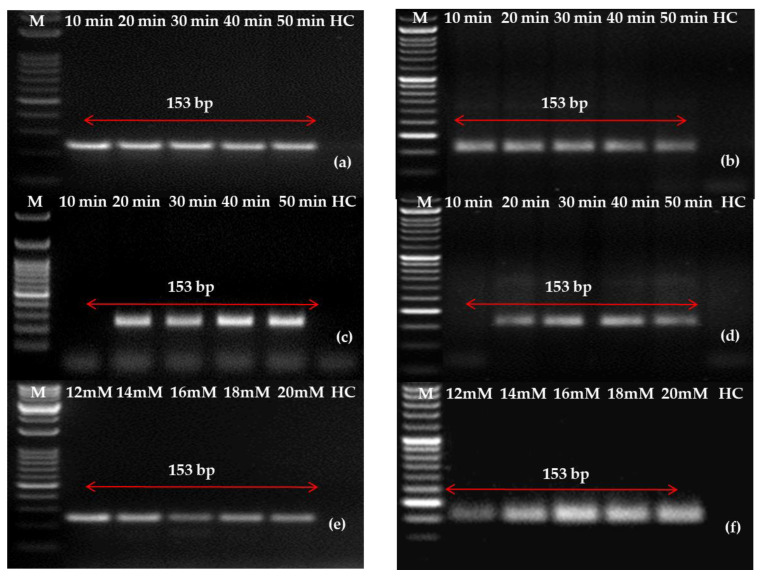
Reverse transcription-recombinase polymerase amplification (RT-RPA) optimization for potato leafroll virus (PLRV) detection using forward and reverse primer LRRPAF3/R3-Set C. The RT-RPA amplified products on (**a**) a PCR machine and (**b**) a water bath using a cDNA template at a temperature of 40 °C and a duration range of 10, 20, 30, 40 and 50 min. Likewise, the RT-RPA amplified amplicons using total RNA (1 μL) in (**c**) a PCR machine and (**d**) a water bath under the same conditions of temperature and duration. Optimization of magnesium acetate concentrations ranging from 12 mM to 20 mM under a similar setup of (**e**) thermal cycler and (**f**) water bath. Lane M: 50 bp ladder; HC: healthy control.

**Figure 6 ijms-24-02511-f006:**
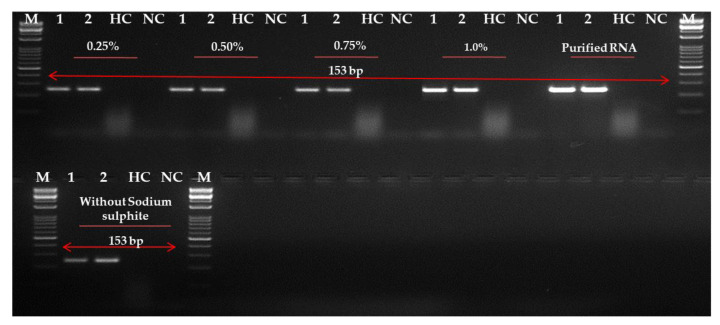
One-step reverse transcription-recombinase polymerase amplification (RT-RPA) optimization for potato leafroll virus (PLRV) detection using forward and reverse primer LRRPAF3/R3-Set C. Utilization of two RNA isolation methods, viz., cellular disc paper-technique [28] and Spectrum^TM^ Plant Total RNA kit (Sigma-Aldrich, St. Louis, MO, USA). The agarose gel images visualize the amplicons when the respective concentration of sodium sulphite ranging from 0.25 to 1.0% for the respective method of RNA extraction has been used. HC: healthy potato mini tuber sample, NC: negative control, Lane M: 50 bp marker.

**Figure 7 ijms-24-02511-f007:**
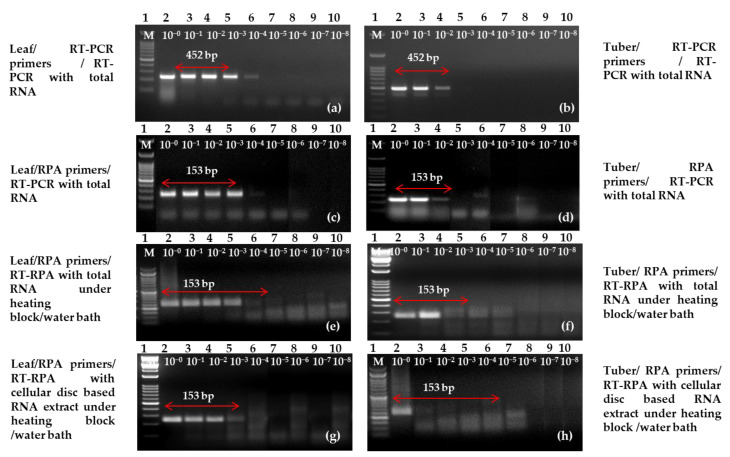
Reverse transcription-polymerase chain reaction (RT-PCR) and reverse transcription-recombinase polymerase amplification (RT-RPA) amplified amplicons at dilutions levels ranging from 10^−0^ to 10^−8^ detect potato leafroll virus (PLRV). One-step RT-PCR with previously reported primers [12] on purified RNA performed with (**a**) infected leaves and (**b**) infected tubers. One-step RT-PCR with optimized primer LRRPAF3/R3 on purified RNA in (**c**) infected leaves and (**d**) infected tubers. One-step RT-RPA with newly optimized primer LRRPAF3/R3 on purified total RNA in (**e**) infected leaves and (**f**) infected tubers. One-step RT-RPA with optimized primer LRRPAF3/R3 on cellular disc-mediated RNA extract in (**g**) infected leaves and (**h**) infected tubers. Lane M (**a**–**h**): 50 bp DNA ladder.

**Figure 8 ijms-24-02511-f008:**
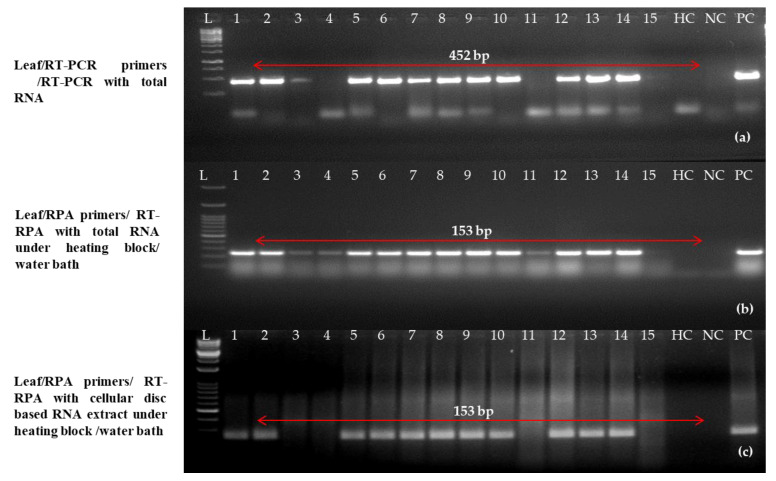
Validation of reverse-transcription recombinase polymerase amplification (RT-RPA) and reverse transcription-polymerase chain reaction (RT-PCR) in one step for potato leafroll virus (PLRV) detection (leaves). (**a**) One-step RT-PCR performed with previously reported primers [12] on purified RNA; (**b**) one-step RT-RPA with newly optimized primer LRRPAF3/R3 on purified total RNA; (**c**) one-step RT-RPA with optimized primer LRRPAF3/R3 on cellular disc-mediated RNA extract. Lane M (**a**–**c**): 50 bp DNA ladder, lanes 1–15 show potato leaf samples, PC: known positive, HC: healthy, NC: negative (water).

**Figure 9 ijms-24-02511-f009:**
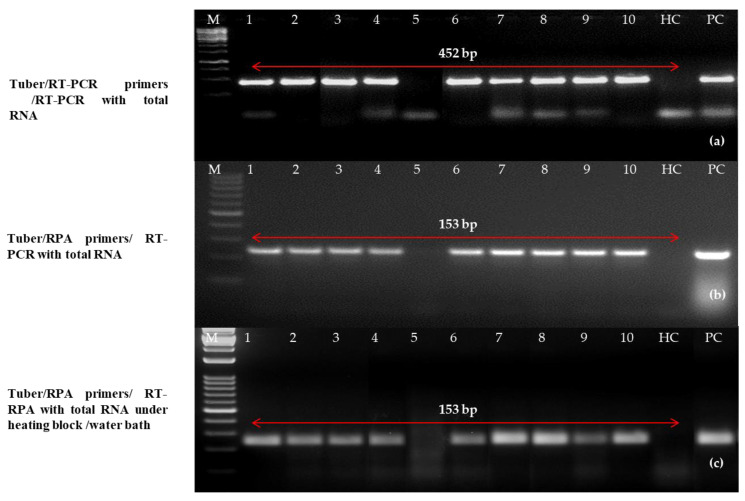
Optimization of reverse-transcription recombinase polymerase amplification/reverse transcription-polymerase chain reaction (RT-RPA/RT-PCR) for successful detection in potato leafroll virus (PLRV) infected tubers. (**a**) One-step RT-PCR performed with previously reported primers [12] on purified RNA; (**b**) one-step RT-RPA with newly optimized primer LRRPAF3/R3 on purified total RNA; (**c**) one-step RT-RPA with optimized primer LRRPAF3/R3 on cellular disc-mediated RNA extract. Lane M (**a**–**c**): 50 bp DNA ladder, lanes 1–15: potato leaf samples, PC: known positive, HC: healthy, NC; negative (water). In all three gel images, a 50 bp DNA marker (Lane M) has been used. The amplicons of infected tuber samples represent lane 1–10 along with a positive control (PC) and a healthy control (HC).

**Table 1 ijms-24-02511-t001:** Primers utilized in this research study with the aim of optimizing a one-step reverse transcription-recombinase polymerase amplification (RT-RPA) technique.

Target Virus	Primer Set No.	Primers	Polarity	Sequence 5′–3′	Bases	Target Position on Coat Protein Gene	Position on Accession Number in NCBI GeneBank (NC_001747.1)	GC (%)	Tm (°C)	Amplicon Size (bp)
PLRV	A	LRRPAF1	Sense	GTATAAGATCACAAGCATCTTACTTCAGTTC	31	327–357	4019–4267	35.5	61.3	249
		LRRPAR1	Antisense	GTATCTGAAGATTTTCCATTTCCCTTC	27	548–575		37	62.5	
	B	LRRPAF2	Sense	TAAGATCACAAGCATCTTACTTCAGTTC	28	330–357	4022–4267	35.7	60.5	246
		LRRPAR2	Antisense	GTATCTGAAGATTTTCCATTTCCCTTC	27	548–575		37	62.5	
	C	LRRPAF3	Sense	ACTCAAGGCCTACCATGAGTATAAGAT	27	309–335	4001–4183	40.7	60.9	153 *
		LRRPAR3	Antisense	GTAATTTGGAACTTGTTGACGTAGGACT	28	434–461		39.3	63	

* The best selected for optimization and validation of one-step RT-RPA.

**Table 2 ijms-24-02511-t002:** Comparison and validation of potato leafroll virus (PLRV) detection methods, viz., one-step reverse-transcription recombinase polymerase amplification (RT-RPA) and reverse transcription-polymerase chain reaction (RT-PCR) of field-collected suspected plant parts of widely cultivated and popular potato cultivars across different agroecological zones in various states of India.

Field Sample No.	One-Step RT-PCR [12]	One Step RT-RPA with Primer Pair C (LRRPAF3/R3) under Heating Block/Water Bath	One Step RT-RPA with Primer Pair C (LRRPAF3/R3) under Heating Block/Water Bath
Leaves	Purified RNA	Purified RNA	Crude RNA (Cellular Disc-Mediated RNA Extract with Added 1.0% Sodium Sulphite)
1	+	+	+
2	+	+	+
3	+	+	−
4	−	−	−
5	+	+	+
6	+	+	+
7	+	+	+
8	+	+	+
9	+	+	+
10	+	+	+
11	−	−	−
12	+	+	+
13	+	+	+
14	+	+	+
15	−	−	−
Healthy Control	−	−	−
Negative Control	−	−	−
Positive Control	+	+	+
Field Sample No.	One-step RT-PCR [12]	One-step RT-RPA with primer pair C (LRRPAF3/R3) under thermal Cycler	One step RT-RPA with primer pair C (LRRPAF3/R3) under heating block/water bath
Tubers	Purified RNA	Purified RNA	Purified RNA
1	+	+	+
2	+	+	+
3	+	+	+
4	+	+	−
5	−	−	−
6	+	+	+
7	+	+	+
8	+	+	+
9	+	+	+
10	+	+	+
Healthy Control	−	−	−
Positive Control	+	+	+

(+): PLRV positive; (−): PLRV negative.

**Table 3 ijms-24-02511-t003:** Comparative performance of double antibody sandwich enzyme-linked immunosorbent assay (DAS-ELISA), reverse transcription-polymerase chain reaction (RT-PCR) and reverse transcription-recombinase polymerase amplification (RT-RPA) methods for potato leafroll virus (PLRV) detection in field-collected suspected plant parts of widely cultivated and popular potato cultivars across different agroecological zones in various states in India.

Sample No.	Location/State	Cultivar/Variety	DAS-ELISA			One-Step RT-PCR [12]		One Step RT-RPA			
						RNA		RNA		Crude RNA	
			Leaves	Dormant Tubers	Sprouted Tubers	Leaves	Dormant Tubers	Leaves	Dormant Tubers	Leaves	Dormant Tubers
1	Gujarat	Kufri Pukhraj	−	−	−	−	−	+	−	+	−
2	Gujarat	Kufri Jyoti	+	−	−	+	+	+	+	+	+
3	Punjab	Kufri Pukhraj	−	−	−	−	−	+	−	+	−
4	Punjab	Kufri Badshah	−	−	−	+	+	+	+	+	−
5	Haryana	Kufri Chandramukhi	+	−	−	+	+	+	+	+	+
6	Haryana	Kufri Jyoti	+	−	+	+	+	+	+	+	+
7	Haryana	Kufri Bahar	−	−	−	−	−	+	−	+	−
8	Himachal Pradesh	Kufri Jyoti	−	−	−	−	−	+	−	+	−
9	Himachal Pradesh	Kufri Himalini	−	−	−	−	−	+	−	+	−
10	Himachal Pradesh	Kufri Chandramukhi	+	−	−	+	+	+	+	+	+
11	Uttar Pradesh	Kufri Pukhraj	+	−	−	+	−	+	−	+	+
12	Uttar Pradesh	Kufri Chipsona 1	+	−	+	+	+	+	+	+	+
13	Uttar Pradesh	Kufri Chipsona 4	−	−	−	+	−	+	−	+	−
14	Uttar Pradesh	Kufri Bahar	−	−	−	+	−	+	+	+	+
15	Bihar	Kufri Sindhuri	−	−	−	−	−	+	+	+	−
16	Bihar	Kufri Chandramukhi	−	−	−	+	−	+	+	+	−
17	Bihar	Kufri Anand	−	−	−	−	−	−	−	−	−
18	Madhya Pradesh	Kufri Pukhraj	+	−	+	+	+	+	+	+	+
19	Madhya Pradesh	Kufri Lauvkar	−	−	−	−	−	+	−	+	−
20	Madhya Pradesh	Kufri Chandramukhi	−	−	−	−	−	+	−	+	−
21	West Bengal	Kufri Jyoti	+	−	−	+	+	+	+	+	+
22	West Bengal	Kufri Chandramukhi	−	−	−	−	−	+	+	+	−
23	West Bengal	Kufri Pukhraj	+	−	−	+	+	+	+	+	+
24	West Bengal	Kufri Ashoka	+	−	+	+	+	+	+	+	+
25	Meghalaya	Kufri Megha	+	−	−	+	+	+	+	+	+
26	Meghalaya	Kufri Giriraj	−	−	−	−	−	+	−	+	−
27	Meghalaya	Kufri Jyoti	+	−	−	+	+	+	+	+	+
Total			13	0	04	16	12	26	16	26	13

(+): PLRV positive; (−): PLRV negative.

## Data Availability

All data generated or analysed during this study are included in this published article and its Appendix A.

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
