# Peer review of "Development of Reverse Transcription Recombinase Polymerase Amplification (RT-RPA): A Methodology for Quick Diagnosis of Potato Leafroll Viral Disease in Potato"

_ijms, 2023, doi:10.3390/ijms24032511_

Round 1

Reviewer 1 Report

This work on reverse transcription recombinase polymerase amplification (RT-RPA) to detect PLRV from naturally infected diverse potato genotypes has been meticulously performed. The authors have standardized this protocol in leaves and tubers which is a first report. The PLRV is a phloem limited virus and so far, the detection methods are limited and resource intensive for routine screening of large number of samples. The simple RNA extraction methodology will be useful in resource limited laboratories. However, I have some suggestions and queries for further improvement of this manuscript

1. Why there is an evident difference in detection of PLRV in dormant and sprouted tubers (Table 3). Please include the reason behind this finding in the discussion part.

2. Why cellular disc paper-based method of RNA extraction worked for leaves, not tubers?

3. Figure 2 require modification in name of aphid vector instead of “winged aphid”. Concise the symptoms details provided in the text box. Give appropriate numbering to sub-figures for better clarity in flow diagram.

4. It is suggested to incorporate the additional relevant and latest references in the introduction and discussion part.

5. Line 367-369 “Because seed tubers are the primary means by which viruses are disseminated, the fact that the established RT-RPA may be used to tubers confers further value on this work.” Please rephrase. 

6. Line 393-394 “In the course of our research, we decided to focus on the coat protein (CP) gene of PLRV due to the prevalence of this gene in the diseased potato”. Please modify the sentence. 

7. Improve the English in introduction and discussion part.

Reviewer 2 Report

This report describes Development of reverse transcription recombinase polymerase 2 amplification (RT-RPA): A methodology for quick diagnosis of potato leafroll viral disease in potato. Please accept the MS included with the following suggestions and corrections.

1.     The authors should explain the length of the primers, according to the RPA protocol for primers design is a minimum of 30 bp maximum of 35 bp. Please add the primer's design reference.

2.     The authors should perform an assay’s limit of copy number detection using the serially diluted cloned plasmid (target) as a template in RPA for validation primers sensitivity and limit of copy number.

3.     Please clarify, whether if tested the current method efficacy using any aphids RNA or asymptomatic samples for detection of PLRV by RT-RPA.

4.     The authors should explain the complications of plant crude sap extraction using GEB buffer, GEB buffer-based sap preparation is commonly used in RPA for plant pathogen detection from the direct plant. Also, what is the difference between the GEB protocol with your current method?

5.     The authors did not perform the primers specificity test using other potato-infecting viruses. Please do it and add these results in the appropriate sections.

6.     A comparison table with includes sensitivity, cost-effectiveness, and time among RT-PCR and conventional RT-RPA is required, and add the table in the supplementary section.

Minor corrections

Line 58: add the additional reference, 10.1016/j.pmpp.2017.03.001

Line 146: Fig 1 can be moved into supplementary fig

Line 175: Fig 3 inside legends move to downside gel image (A-219 bp, B 246 bp, etc.,)

Line 201: fig 4 a change better one

Mark all DNA ladders

Include keywords: Isothermal and PLRV

Round 2
